# Alcohol Participates in the Synthesis of Functionalized Coumarin-Fused Pyrazolo[3,4-*b*]Pyridine from a One-Pot Three-Component Reaction

**DOI:** 10.3390/molecules24152835

**Published:** 2019-08-04

**Authors:** Wei Lin, Cangwei Zhuang, Xiuxiu Hu, Juanjuan Zhang, Juxian Wang

**Affiliations:** 1School of Chemistry and Environmental Engineering, Jiangsu University of Technology, Changzhou 213001, China; 2State Key Laboratory of Pharmaceutical Biotechnology, Nanjing University, Nanjing 210093, China; 3Institute of Medicinal Biotechnology, Chinese Academy of Medical Science and Peking Union Medical College, Beijing 100050, China

**Keywords:** coumarin, pyrazolo[3,4-*b*]pyridine, synthesis, silica sulfuric acid

## Abstract

A concise and efficient approach to synthesizing coumarin-fused pyrazolo[3,4-*b*]pyridine via silica sulfuric acid (SSA) catalyzed three-component domino reaction under microwave irradiation has been demonstrated. Participation of various alcohols in construction of coumarin derivatives has been described for the first time. Short reaction time, high yields, one-pot procedure, usage of eco-friendly catalyst, and solvent are the key features of this method.

## 1. Introduction

As one of the most important heterocyclic compounds, coumarin was widely found in nature products [1,2], and several synthetic coumarins [3] with a variety of pharmacophoric groups at C-3, C-4, and C-7 positions have been intensively screened for various biological activities like AChE inhibitors [4,5,6], anticancer [7,8,9], anticoagulant [10,11], anti-HIV [12,13,14], antitubercular [15,16], anti-inflammatory [17,18], antioxidant [19], antibacterial [20], antihypertensive [21], anticonvulsant [22], antifungal [23], and antihyperglycemic [24]. A recent literature survey suggests quite a few coumarin derivatives have been patented for their biological properties (Figure 1). Besides the high biological activity, coumarin is also considered to be a functional material [25,26] such as receptors [27,28,29], signaling units in sensors and biosensors, as well as in advanced photophysical systems [30,31].

Among various nitrogen-containing heterocyclic compounds, pyrazolo[3,4-*b*]pyridine is recognized as important drug molecular skeleton in recent years due to a wide varieties of biological activities (Figure 2), such as antimicrobial [32,33], anti-inflammatory [34,35], anti-proliferative [36,37], and many other [38,39] important effects.

Therefore, development and introduction of a convenient, efficient method for the synthesis of coumarin-fused pyrazolo[3,4-*b*]pyridine is highly desirable for their immense pharmacological potential. As a part of our research on the synthesis of novel functionalized heterocyclic derivatives [40,41,42,43,44,45,46], in the current paper, we report a novel three-component domino reaction for the synthesis of functionalized coumarin-fused pyrazolo[3,4-*b*]pyridine derivatives using silica sulfuric acid as the catalyst. It worth mentioning that participation of alcohols in construction of coumarin derivatives is described for the first time.

## 2. Results and Discussion

In the early literature reports of our group [44], the coumarino[4,3-d]pyrazolo[3,4-b]pyridine derivative (**3a**) was synthesized by the reaction of 3-acylcoumarin (**1a**) with 5-aminopyrazole (**2a**) catalyzed by silica sulfuric acid (SSA) in EtOH at 90 °C for 20 min under microwave irradiation (Scheme 1).

According to our previously reported synthetic procedure, we speculate that the coumarin derivative **6a** could be obtained from the 2-butyryl-3*H*-benzo[*f*]chromen-3-one (**4a**) and 3-methyl-1-phenyl-1*H*-pyrazol-5-amine (**2a**) used as the starting materials. However, product **6a** was not available as expect (Scheme 2-1). Considering the steric hindrance effect of the reaction, when ethanol and ethylene glycol (EG) as mixed solvent (volume ratio of EG/EtOH = 1:1) was added to the reaction, and further increasing the temperature (120 °C), a new product **7a** formed unexpectedly (Scheme 2-2), which was identified by ^1^H-NMR, ^13^C-NMR, HRMS analysis. Moreover, we also obtained the single crystal of **7a** suitable for X-ray analysis (Figure 3) [47]. To our surprise, the solvent ethanol also participated in this reaction and a novel coumarin derivative was constructed.

In order to achieve the optimal conditions of three-component reaction, a series of catalysts, solvents, and temperature were screened, as shown in Table 1. Some other acid catalysts such as *p*-TsOH, HClO_3_S, H_2_SO_4_, SiO_2_-H_2_SO_4_ (Table 1. entries 1, 3–5) and base catalysts such as K_2_CO_3_, NaOH, Cs_2_CO_3_ (Table 1, entries 6–8) were tested. However, none of them gave better results, lead to the identification of SSA as the most effective catalyst (Table 1. entry 2). To further increase the yield of desired product **7a**, different solvents were evaluated. The results revealed that EtOH and EG as mixed solvents greatly improved the transformation, in control to EtOH, PEG, glycerol, and DMF as a single solvent (Table 1, entries 2, 9–12). When the volume ratio of EG/EtOH = 3:1, the yield of **7a** could further increase to 68% (Table 1, entry 15). Much to our delight, we observed that increasing of the temperature to 140 °C resulted in affording **7a** in 84% yield (Table 1, entry 20).

With optimal conditions in hand, the corresponding novel coumarin-fused pyrazolo[3,4-*b*]pyridine derivatives **7** were synthesized (Scheme 3).

As illustrated in Scheme 3, the substrate scope of the transformation was examined using arylbenzo[f]chromen-3-one **4**, enaminone **2**, and alkyl alcohol **5** as staring materials. Notably, electronic effects had an important impact on this reaction. When the substituent R^3^ was electron-donating group, such as Me, the desired products could not be obtained at all (**7e**, **7f**).

To further expand the scope of substrates, aryl alcohols (**8**) instead of alkyl alcohols (**5**) were also tested. It was found that aryl alcohols were well tolerated under the optimal reaction conditions, the corresponding products were afforded in moderate to good yields. When substituent R^3^ was electron-withdrawing groups (Ph), the yields were good and no more than 1 h cost (Table 2, entries 1–19). However, the substituents R^3^ was electron-donating groups (CH_3_) (Table 2, entries 20–22), the yields were lower and the reaction time was longer. Unfortunately, When R^3^ and R^4^ was electron rich group, such as Me, the reaction could not proceed successfully (Table 2, entry 23).

To gain insight into the mechanism of this one-spot three-component reaction process, some additional experiments were performed. When benzaldehyde (**10**) was added to the reaction instead of phenylmethanol (**8a**) under standard conditions, 73% yield of desired product (**9a**) could be obtained, and reaction time reduced from 1 h to 15 min (Scheme 4A), and when butyraldehyde (**11**) was added to the reaction 50% yield of desired product (**7c**) could be obtained (Scheme 4B). The reaction did not proceed successfully without SSA catalyzed. Just phenylmethanol (**8a**) was heated to 140 °C, directly with the catalyst of SSA, benzaldehyde (**10**) and benzoic acid (**12**) could be detected by GC-MS (Scheme 4C). We speculated that the benzaldehyde was most likely the key intermediate in this protocol.

Herein, we propose the following mechanism for the reaction (Scheme 5). SSA catalyzed alkyl alcohol **5** to afford the corresponding aldehyde, then the intermediate **A** is formed by means of a Knoevenagel condensation of aldehyde and arylbenzo[*f*]chromen-3-one (**4**). The intermediate **A** is activated by SSA, which subsequently undergoes Michael addition with enaminone (**2**) via attack of the nucleophilic C-4 of the intermediate **A** to give intermediate **B**, which transformed to more-stable intermediate **C**. Then, intermediate **C** tautomerizes to intermediate **D**, which undergoes intramolecular nucleophilic addition to form intermediate **E**. In the last step, loss of H_2_O affords the desired product.

## 3. Conclusions

In conclusion, we have developed a protocol for the facile synthesis of various potentially biologically active coumarin-fused pyrazolo[3,4-*b*]pyridine derivatives, based on a novel three-component domino reaction under microwave irradiation. Using this method, coumarin derivatives could be rapidly constructed in moderate-to-good yields with short reaction time. Further study to deeply understand the reaction mechanism is currently underway in our lab.

## 4. Experimental Section

### 4.1. General

All reagents were purchased from commercial suppliers (Aladdin, Shanghai, China) and used without further purification. Microwave irradiation was carried out with Initiator 2.5 Microwave Synthesizers from Biotage, Uppsala, Sweden. The reaction temperatures were measured by infrared detector during microwave heating. Melting points are uncorrected. IR spectra were recorded on a Tensor 27 spectrometer (Bruker Corp., Karlsruhe, Germany) in KBr with absorptions in cm^−1^. ^1^H-NMR (400 MHz) and ^13^C-NMR (75 MHz or 100 MHz) spectra were recorded on a Varian Inova-400 MHz or Varian Inova-300 MHz (Varian, CA, America) in CDCl_3_, DMSO-*d*_6_ or CF_3_COOD as solution. *J* values are in hertz. Chemical shifts are expressed in parts per million downfield from interal standard TMS. High-resolution mass spectra (HRMS) for all the compounds were determined on Bruker MicrOTOF-QII mass spectrometer (Bruker Corp., Karlsruhe, Germany) with ESI resource. X-ray diffraction analysis was recorded on a Smart-1000 diffractometer (PANalytical B.V., Holland).

### 4.2. General Procedure for the Synthesis of Products 4 Are Represented as Follows

Typically, 2-hydroxy-1-naphthaldehyde (5 mmol), ethyl 3-oxopentanoate or ethyl 3-oxohexanoate or ethyl acetoacetate (5 mmol) and piperidine (0.5 mmol) were introduced in a 20 mL vial with ethanol (10 mL) as solution. Subsequently, the reaction vial was closed and then prestirred for 10 s. The mixture was irradiated at 90 °C for 10 min. After the completion, the reaction mixture was then cooled to room temperature and concentrated in vacuo to remove the solvent. The residue was then washed with water, filtered, dried, and the precipitate was purified by recrystallization from 95% EtOH to give the products of **4**. The analytical data for represent compounds are shown below. ^1^H-NMR and ^13^C-NMR spectra of compounds **4** in Appendix A.

#### 4.2.1. *2-Butyryl-3H-benzo[f]chromen-3-one* (**4a**)

Yellow solid; yield 89%; m.p.: 127–129 °C; IR (KBr): *ν* 1734, 1626, 1557, 1513, 1383, 1109, 864 cm^−1^; ^1^H-NMR (CDCl_3_, 400 MHz) *δ* (ppm): 9.21 (s, 1H, ArH), 8.91 (s, 1H, ArH), 8.06 (d, *J* = 8.8 Hz, 1H, ArH), 7.83 (d, *J* = 8.8 Hz, 1H, ArH), 7.57–7.56 (m, 1H, ArH), 7.20 (d, *J* = 9.2 Hz, 1H, ArH), 7.12 (dd, *J*_1_ = 8.8 Hz, *J*_2_ = 2.0 Hz, 1H, ArH), 3.00 (t, *J* = 7.2 Hz, 2H, CH_2_), 1.63–1.58 (m, 2H, CH_2_), 0.93 (t, *J* = 7.2 Hz, 3H, CH_3_); ^13^C-NMR (100 MHz, DMSO-*d*_6_) *δ* (ppm): 197.5, 159.1, 158.9, 156.4, 142.8, 136.6, 132.1, 131.5, 124.6, 121.9, 118.6, 113.0, 111.4, 104.8, 43.9, 17.3, 14.1;

#### 4.2.2. *2-Butyryl-9-methoxy-3H-benzo[f]chromen-3-one* (**4b**)

Yellow solid, yield 88%; m.p.: 125–128 °C; IR (KBr) *ν*: 1730, 1667, 1601, 1556, 1513, 1386, 1365, 1196, 948, 836 cm^−1^; ^1^H-NMR (CDCl_3_, 400 MHz) *δ* (ppm): 9.01 (s, 1H, ArH), 7.86 (d, *J* = 8.8 Hz, 1H, ArH), 7.68 (d, *J* = 8.8 Hz, 1H, ArH), 7.40 (s, 1H, ArH), 7.14 (t, *J* = 8.4 Hz, 2H, ArH), 3.94 (s, 3H, CH_3_O), 3.12 (t, *J* = 8.4 Hz, 2H, CH_2_); 1.76–1.70 (m, 2H, CH_2_), 1.00 (t, *J* = 8.4 Hz, 3H, CH_3_); ^13^C-NMR (100 MHz, CDCl_3_) *δ* (ppm): 197.6, 159.9, 158.7, 156.0, 142.5, 135.3, 131.2, 130.2, 124.7, 121.1, 117.9, 113.1, 111.4, 100.7, 55.5, 44.0, 16.9, 13.3.

#### 4.2.3. *2-Propionyl-3H-benzo[f]chromen-3-one* (**4c**)

Yellow solid, yield 87%; m.p.: 134–136 °C; IR (KBr): *ν* 1732, 1662, 1601, 1556, 1524, 1387, 1365, 1196, 945, 823 cm^−1^; ^1^H-NMR (CDCl_3_, 400 MHz) *δ* (ppm): 9.15 (s, 1H, ArH), 8.74 (s, 1H, ArH), 7.90 (d, *J* = 8.8 Hz, 1H, ArH), 7.66 (d, *J* = 8.8 Hz, 1H, ArH), 7.41–7.40 (m, 1H, ArH), 7.04 (d, *J* = 8.8 Hz, 1H, ArH), 6.95 (dd, *J*_1_ = 8.8 Hz, *J*_2_ = 2.0 Hz, 1H, ArH), 3.14–3.08 (m, 2H, CH_2_), 1.08 (t, *J* = 7.2 Hz, 3H, CH_3_); ^13^C-NMR (100 MHz, CDCl_3_) *δ* (ppm): 198.4, 159.9, 159.4, 156.7, 143.3, 135.9, 131.9, 130.8, 125.3, 121.8, 118.7, 113.7, 112.1, 102.0, 35.2, 10.7.

#### 4.2.4. *9-Methoxy-2-propionyl-3H-benzo[f]chromen-3-one* (**4d**)

Yellow solid, yield 87%; m.p.: 125–128 °C; IR (KBr): *ν* 1730, 1667, 1601, 1556, 1513, 1386, 1365, 1196, 948, 836 cm^−1^; ^1^H-NMR (DMSO-*d*_6_, 400 MHz) *δ* (ppm): 9.19 (s, 1H, ArH), 8.14 (d, *J =* 9.2 Hz, 1H, ArH), 7.90 (d, *J =* 9.2 Hz, 1H, ArH), 7.79 (s, 1H, ArH), 7.32 (t, *J =* 8.8 Hz, 1H, ArH), 7.21 (dd, *J_1_ =* 8.8 Hz, *J_2_ =* 2.0 Hz, 1H, ArH), 3.98 (s, 3H, CH_3_O), 3.10–3.05 (m, 2H, CH_2_), 1.09 (t, *J =* 7.2 Hz, 3H, CH_3_); ^13^C-NMR (100 MHz, DMSO-*d*_6_) *δ* (ppm): 198.7, 160.4, 158.8, 156.1, 143.0, 136.2, 131.9, 131.2, 125.4, 122.8, 118.6, 113.9, 112.1, 102.4, 56.2, 35.4, 8.4.

#### 4.2.5. *2-Acetyl-3H-benzo[f]chromen-3-one* (**4e**)

Yellow solid, yield 88%; m.p.: 189–190 °C [48]; IR (KBr): *ν* 2959, 1696, 1622, 1562, 1384, 1227, 1206, 857 cm^−1^.

### 4.3. General Procedure for the Synthesis of Products ***7*** and ***9*** Are Represented as Follows

Typically, benzo[*f*]chromen-3-one **4** (0.5 mmol), enaminone **2** (0.5 mmol), alkyl alcohol **5** (1.0 mL) or aryl alcohols **8** (1.0 mL) and SSA (0.25 g) were introduced in a 5 mL vial with ethylene glycol (3 mL) as solution. Subsequently, the reaction vial was closed and then prestirred for 10 s. The mixture was irradiated at 140 °C. The reaction was monitored by TLC. After the completion, the reaction mixture was then cooled to room temperature and diluted with cold water (30 mL), and extracted with CH_2_Cl_2_ (3 × 30 mL). The extracts were washed with water (3 × 50 mL) and dried over anhydrous Na_2_SO_4_. After evaporation of the solvent under reduced pressure, the precipitate was collected and purified by recrystallization from 95% EtOH or by flash column chromatography (petroleum ether:ethyl acetate = 8:1) to give the products **7** or **9**. The analytical data for represent compounds are shown below. ^1^H-NMR and ^13^C-NMR spectra of compounds **7** and **9** in Appendix A.

#### 4.3.1. *2-(5-Ethyl-3,4-dimethyl-1-phenyl-1H-pyrazolo[3,4-b]pyridin-6-yl)-3H-benzo[f]chromen-3-one* (**7a**)

White solid, m.p.: 258–260 °C; IR (KBr, cm^−1^) *ν*: 2960, 1722, 1629, 1572, 1507, 1415, 1387, 1315, 1290, 1248, 1211, 1096, 989, 906, 815, 787, 713, 691, 605; ^1^H-NMR (400 MHz, DMSO-*d*_6_) *δ* (ppm): 9.07 (s, 1H, ArH), 8.58 (d, *J* = 8.0 Hz, 1H, ArH), 8.23 (t, *J* = 8.0 Hz, 3H, ArH), 8.08 (d, *J* = 8.0 Hz, 1H, ArH), 7.69–7.61 (m, 3H, ArH), 7.44 (t, *J* = 8.0 Hz, 2H, ArH), 7.20 (t, *J* = 7.2 Hz, 1H, ArH), 2.78–2.66 (m, 8H, 2 × CH_3_ + CH_2_), 1.05 (t, *J* = 7.2 Hz, 3H, CH_3_); ^13^C-NMR (75 MHz, CF_3_COOD) *δ* (ppm): 156.0, 148.8, 146.2, 145.4, 140.5, 139.1, 135.6, 134.1, 132.6, 132.0, 131.2, 130.8, 130.2, 129.4, 128.5, 126.5, 122.3, 121.3, 116.7, 116.2, 22.7, 17.0, 14.3, 13.4; HRMS: *m*/*z* cacld. for C_29_H_24_N_3_O_2_ [M + H]^+^ 446.1869, Found 446.1853.

#### 4.3.2. *2-(4,5-Diethyl-3-methyl-1-phenyl-1H-pyrazolo[3,4-b]pyridin-6-yl)-3H-benzo[f]chromen-3-one* (**7b**)

White solid, m.p.: >300 °C; IR (KBr, cm^−1^) *ν*: 2974, 1719, 1688, 1656, 1628, 1596, 1628, 1571, 1546, 1506, 1413, 1357, 1204, 1071, 909, 817, 752, 694, 676, 589; ^1^H-NMR (400 MHz, DMSO-*d*_6_) *δ* (ppm): 9.16 (s, 1H, ArH), 8.66 (d, *J* = 8.4 Hz, 1H, ArH), 8.29 (d, *J* = 9.2 Hz, 1H, ArH), 8.21 (d, *J* = 7.6 Hz, 2H, ArH), 8.11 (d, *J* = 8.0 Hz, 1H, ArH), 7.73–7.63 (m, 3H, ArH), 7.47 (t, *J* = 8.0 Hz, 2H, ArH), 7.23 (t, *J* = 7.6 Hz, 1H, ArH), 3.51–3.48 (m, 2H, CH_2_), 3.17–3.14 (m, 2H, CH_2_), 2.79 (s, 3H, CH_3_), 1.35 (t, *J* = 7.2 Hz, 3H, CH_3_), 1.09 (t, *J* = 7.6 Hz, 3H, CH_3_); ^13^C-NMR (75 MHz, DMSO-*d*_6_) *δ* (ppm): 160.2, 154.3, 153.8, 149.4, 148.1, 142.4, 139.6, 134.1, 130.5, 130.3, 129.6, 129.0, 128.3, 126.8, 125.7, 123.2, 120.5, 117.2, 115.9, 113.5, 100.0, 22.2, 21.6, 16.6, 16.2, 15.5; HRMS: *m*/*z* cacld. for C_30_H_25_N_3_O_2_ (M)^+^ 459.1947, Found 459.1946.

#### 4.3.3. *2-(5-Ethyl-3-methyl-1-phenyl-4-propyl-1H-pyrazolo[3,4-b]pyridin-6-yl)-3H-benzo[f]chromen-3-one* (**7c**)

White solid, m.p.: 242–245 °C; IR (KBr, cm^−1^) *ν*: 2974, 2880, 2703, 2545, 1789, 1722, 1665, 1573, 1503, 1439, 1414, 1389, 1359, 1320, 1288, 1248, 1217, 1155, 1091, 915, 858, 813, 792, 745, 695, 641, 610; ^1^H-NMR (400 MHz, DMSO-*d*_6_) *δ* (ppm): 9.14 (s, 1H, ArH), 8.63 (d, *J* = 8.8 Hz, 1H, ArH), 8.26 (d, *J* = 8.8 Hz, 1H, ArH), 8.22 (d, *J* = 8.0 Hz, 2H, ArH), 8.09 (d, *J* = 8.0 Hz, 1H, ArH), 7.70–7.63 (m, 3H, ArH), 7.46 (t, *J* = 7.6 Hz, 2H, ArH), 7.22 (t, *J* = 7.6 Hz, 1H, ArH), 3.07–3.06 (m, 2H, CH_2_), 2.76–2.73 (m, 5H, CH_3_ + CH_2_), 1.72–1.69 (m, 2H, CH_2_), 1.13 (s, 3H, CH_3_), 1.07 (s, 3H, CH_3_); ^13^C-NMR (75 MHz, CF_3_COOD) *δ* (ppm): 165.7, 155.9, 148.1, 146.2, 145.9, 140.8, 139.1, 135.0, 134.1, 132.6, 131.9, 131.2, 130.7, 130.2, 129.3, 128.5, 126.5, 121.6, 121.3, 116.3, 33.4, 26.2, 22.2, 14.5, 13.7; HRMS: *m*/*z* cacld. for C_31_H_28_N_3_O_2_ [M + H]^+^ 474.2182, Found 474.2210.

#### 4.3.4. *2-(5-Ethyl-3,4-dimethyl-1-phenyl-1H-pyrazolo[3,4-b]pyridin-6-yl)-9-methoxy-3H-benzo[f]chromen-3-one* (**7d**)

White solid, m.p.: >300 °C; IR (KBr, cm^−1^) *ν*: 2975, 2026, 1795, 1728, 1628, 1574, 1509, 1230, 1091, 989, 917, 840, 794, 751, 686, 610; ^1^H-NMR (400 MHz, DMSO-*d*_6_) *δ* (ppm): 9.21 (s, 1H, ArH), 8.24–8.17 (m, 3H, ArH), 8.01–7.96 (m, 2H, ArH), 7.50–7.45 (m, 3H, ArH), 7.27–7.21 (m, 2H, ArH), 3.90 (s, 3H, OCH_3_), 2.80–2.79 (m, 8H, CH_2_ + 2 × CH_3_), 1.07 (t, *J* = 7.2 Hz, 3H, CH_3_); ^13^C-NMR (75 MHz, DMSO-*d*_6_) *δ* (ppm): 160.3, 160.1, 154.4, 154.0, 149.0, 143.1, 142.4, 140.0, 139.7, 133.8, 131.4, 131.0, 129.5, 127.5, 125.6, 125.5, 120.3, 118.7, 116.9, 114.3, 102.6, 56.3, 22.4, 16.1, 15.4, 15.0; HRMS: *m*/*z* cacld. for C_30_H_26_N_3_O_3_ [M + H]^+^ 476.1974, Found 476.1980.

#### 4.3.5. *2-(5-Ethyl-3-methyl-1,4-diphenyl-1H-pyrazolo[3,4-b]pyridin-6-yl)-3H-benzo[f]chromen-3-one* (**9a**)

Yellow solid, m.p.: >300 °C; IR (KBr, cm^−1^) *ν*: 3032, 2978, 2888, 2763, 1725, 1049, 958, 815, 756, 699, 679, 588; ^1^H-NMR (400 MHz, CF_3_COOD) *δ* (ppm): 10.16 (s, 1H, ArH), 9.15–9.14 (m, 2H, ArH), 8.88–8.87 (m, 1H, ArH), 8.65–8.64 (m, 1H, ArH), 8.59–8.55 (m, 4H, ArH), 8.51–8.47 (m, 6H, ArH), 8.38–8.37 (m, 2H, ArH), 3.77–3.76 (m, 2H, CH_2_), 3.05 (s, 3H, CH_3_), 1.91 (s, 3H, CH_3_); ^13^C-NMR (75 MHz, CF_3_COOD) *δ* (ppm): 163.7, 156.1, 149.4, 146.8, 146.4, 140.8, 139.1, 135.8, 134.3, 132.8, 132.6, 132.0, 131.6, 131.2, 130.8, 130.2, 129.9, 129.5, 128.5, 127.8, 126.5, 121.9, 121.4, 116.8, 116.3, 113.6, 22.9, 14.3, 12.5; HRMS: *m*/*z* cacld. for C_34_H_26_N_3_O_2_ [M + H]^+^ 508.2025, Found 508.2025.

#### 4.3.6. *2-(5-Ethyl-3-methyl-1-phenyl-4-(p-tolyl)-1H-pyrazolo[3,4-b]pyridin-6-yl)-3H-benzo[f]chromen-3-one* (**9b**)

Yellow solid, m.p.: >300 °C; IR (KBr, cm^−1^) *ν*: 2968, 1972, 1779, 1572, 1505, 1413, 1360, 1207, 1088, 961, 898, 806, 758, 728, 690, 642; ^1^H-NMR (400 MHz, CF_3_COOD) *δ* (ppm): 10.13 (s, 1H, ArH), 9.15–9.09 (m, 2H, ArH), 8.87–8.84 (m, 1H, ArH), 8.63–8.60 (m, 1H, ArH), 8.54–8.40 (m, 9H, ArH), 8.25–8.24 (m, 2H, ArH), 3.76–3.74 (m, 2H, CH_2_), 3.37 (s, 3H, CH_3_), 3.06 (s, 3H, CH_3_), 1.88–1.87 (m, 3H, CH_3_); ^13^C-NMR (75 MHz, CF_3_COOD) *δ* (ppm): 162.8, 155.1, 148.6, 145.6, 145.4, 142.0, 139.8, 138.2, 135.0, 133.3, 131.7, 131.1, 130.3, 129.9, 129.6, 129.3, 128.7, 128.5, 127.6, 126.9, 125.6, 121.0, 120.4, 115.9, 115.4, 112.7, 21.9, 19.5, 13.4, 11.7; HRMS: *m*/*z* cacld. for C_35_H_28_N_3_O_2_ [M + H]^+^ 522.2182, Found 522.2180.

#### 4.3.7. *2-(5-Ethyl-4-(4-methoxyphenyl)-3-methyl-1-phenyl-1H-pyrazolo[3,4-b]pyridin-6-yl)-3H-benzo[f]chromen-3-one* (**9c**)

Yellow solid, m.p.: >300 °C; IR (KBr, cm^−1^) *ν*: 2967, 1711, 1597, 1571, 1505, 1412, 1286, 1249, 1211, 1048, 982, 897, 849, 806, 758, 690, 641, 587; ^1^H-NMR (400 MHz, CF_3_COOD) *δ* (ppm): 9.29 (s, 1H, ArH), 8.28–8.23 (m, 2H, ArH), 7.99 (d, *J* = 8.4 Hz, 1H, ArH), 7.76 (t, *J* = 7.6 Hz, 1H, ArH), 7.69–7.56 (m, 7H, ArH), 7.50 (d, *J* = 8.4 Hz, 2H, ArH), 7.35 (d, *J* = 8.4 Hz, 2H, ArH), 4.06 (s, 3H, OCH_3_), 2.93–2.88 (m, 2H, CH_2_), 2.23 (s, 3H, CH_3_), 1.02 (t, *J* = 7.2 Hz, 3H, CH_3_); ^13^C-NMR (75 MHz, CF_3_COOD) *δ* (ppm): 162.9, 160.9, 160.5, 155.2, 148.4, 145.8, 145.7, 140.0, 138.3, 135.3, 133.5, 131.8, 131.2, 130.4, 130.0, 129.4, 129.2, 128.7, 127.7, 125.8, 125.3, 121.3, 120.6, 116.0, 115.5, 114.9, 112.8, 55.1, 22.1, 13.4, 12.0; HRMS: *m*/*z* cacld. for C_35_H_28_N_3_O_3_ [M + H]^+^ 538.2131, Found 538.2111.

#### 4.3.8. *2-(5-Ethyl-4-(3-methoxyphenyl)-3-methyl-1-phenyl-1H-pyrazolo[3,4-b]pyridin-6-yl)-3H-benzo[f]chromen-3-one* (**9d**)

Yellow solid, m.p.: >300 °C;.IR (KBr, cm^−1^) *ν*: 2965, 2023, 1785, 1712, 1573, 1504, 1382, 1357, 1285, 1158, 1136, 1046, 782, 759, 712, 689, 588; ^1^H-NMR (400 MHz, DMSO-*d*_6_) *δ* (ppm): 9.23 (s, 1H, ArH), 8.66 (d, *J* = 8.4 Hz, 1H, ArH), 8.29–8.23 (m, 3H, ArH), 8.10 (d, *J* = 8.0 Hz, 1H, ArH), 7.73–7.62 (m, 3H, ArH), 7.54–7.47 (m, 3H, ArH), 7.25 (t, *J* = 7.6 Hz, 1H, ArH), 7.14–7.12 (m, 1H, ArH), 7.03–7.01 (m, 2H, ArH), 3.84 (s, 3H, OCH_3_), 2.58–2.56 (m, 2H, CH_2_), 1.96 (s, 3H, CH_3_), 0.89 (t, *J* = 7.2 Hz, 3H, CH_3_); ^13^C-NMR (75 MHz, DMSO-*d*_6_) *δ* (ppm): 160.2, 159.6, 154.2, 153.9, 148.7, 145.3, 142.9, 140.1, 139.6, 137.3, 134.3, 130.5, 130.4, 130.1, 129.6, 129.5, 129.4, 129.0, 127.8, 126.8, 125.9, 123.1, 121.4, 120.6, 117.2, 115.8, 114.6, 113.5, 55.8, 22.5, 16.0, 14.2; HRMS: *m*/*z* cacld. for C_35_H_27_N_3_O_3_ (M)^+^ 537.2052, Found 537.2053.

#### 4.3.9. *2-(4-(4-Bromophenyl)-5-ethyl-3-methyl-1-phenyl-1H-pyrazolo[3,4-b]pyridin-6-yl)-3H-benzo[f]chromen-3-one* (**9e**)

Yellow solid, m.p.: >300 °C;.IR (KBr, cm^−1^) *ν*: 2968, 2032, 1775, 1721, 1574, 1385, 1357, 1285, 1166, 1047, 782, 759, 712, 681, 588; ^1^H-NMR (400 MHz, DMSO-*d*_6_) *δ* (ppm): 10.19 (s, 1H, ArH), 8.57–8.53 (m, 2H, ArH), 8.43 (d, *J* = 9.2 Hz, 1H, ArH), 8.06 (d, *J* = 8.0 Hz, 1H, ArH), 7.86 (t, *J* = 7.6 Hz, 1H, ArH), 7.78–7.23 (m, 10H, ArH), 2.79 (s, 2H, CH_2_), 2.54(s, 3H, CH_3_), 1.35 (t, *J* = 7.2 Hz, 3H, CH_3_); ^13^C-NMR (75 MHz, DMSO-*d*_6_) *δ* (ppm): 165.7, 159.3, 157.7, 152.3, 151.4, 144.7, 141.6, 138.9, 134.8, 132.3, 131.0, 129.7, 126.7, 125.2, 121.4, 118.1, 113.1, 111.4, 111.3, 109.6, 107.5, 21.9, 17.0, 14.6; HRMS: *m*/*z* cacld. for C_34_H_24_BrN_3_O_2_ (M)^+^ 585.1052, Found 585.1057.

#### 4.3.10. *2-(5-Ethyl-3-methyl-1-phenyl-4-(pyridin-4-yl)-1H-pyrazolo[3,4-b]pyridin-6-yl)-3H-benzo[f]chromen-3-one* (**9f**):

Yellow solid, m.p.: >300 °C; IR (KBr, cm^−1^) *ν*: 2965, 1972, 1783, 1573, 1505, 1413, 1362, 1089, 961, 898, 805, 758, 693, 642; ^1^H-NMR (400 MHz, CF_3_COOD) *δ* (ppm): 10.09 (s, 1H, ArH), 9.11–9.05 (m, 2H, ArH), 8.80 (d, *J* = 8.0 Hz, 1H, ArH), 8.60–8.35 (m, 10H, ArH), 8.21–8.19 (m, 2H, ArH), 3.72–3.70 (m, 2H, CH_2_), 3.01 (s, 3H, CH_3_), 1.83 (t, *J* = 6.8 Hz, 3H, CH_3_);^13^C-NMR (75 MHz, CF_3_COOD) *δ* (ppm): 162.8, 155.0, 148.5, 145.5, 145.4, 142.0, 139.7, 138.1, 135.0, 133.3, 131.6, 131.0, 130.2, 129.8, 129.5, 129.2, 128.7, 128.5, 127.5, 126.8, 125.5, 121.0, 120.4, 115.8, 115.3, 112.6, 21.8, 13.3, 11.6; HRMS: *m*/*z* cacld. for C_33_H_25_N_4_O_2_ [M + H]^+^ 509.1978, Found 509.1963.

#### 4.3.11. *2-(5-Ethyl-4-(furan-2-yl)-3-methyl-1-phenyl-1H-pyrazolo[3,4-b]pyridin-6-yl)-3H-benzo[f]chromen-3-one* (**9g**):

Yellow solid, m.p.: >300 °C; IR (KBr, cm^−1^) *ν*: 2966, 1720, 1629, 1566, 1412, 1383, 1264, 1084, 959, 852, 797, 766, 724, 691, 640, 617; ^1^H-NMR (400 MHz, CF_3_COOD) *δ* (ppm): 9.24 (s, 1H, ArH), 8.21 (d, *J* = 9.2 Hz, 1H, ArH), 7.95 (d, *J* = 8.8 Hz, 1H, ArH), 7.65–7.63 (m, 2H, ArH), 7.62–7.55 (m, 6H, ArH), 7.46 (d, *J* = 9.2 Hz, 1H, ArH), 7.39 (m, 3H, ArH), 2.93–2.87 (m, 2H, CH_2_), 2.21 (s, 3H, CH_3_), 1.03 (t, *J* = 7.6 Hz, 3H, CH_3_); ^13^C-NMR (75 MHz, CF_3_COOD) *δ* (ppm): 163.0, 160.2, 156.0, 148.7, 145.7, 142.1, 140.0, 137.9, 135.2, 133.5, 131.9, 131.5, 130.7, 130.5, 129.8, 129.0, 127.1, 126.9, 125.8, 121.2, 118.0, 115.4, 113.7, 112.2, 22.1, 13.6, 11.8; HRMS: *m*/*z* cacld. for C_32_H_24_N_3_O_3_ [M + H]^+^ 498.1818, Found 498.1831.

#### 4.3.12. *2-(5-Ethyl-3-methyl-1,4-diphenyl-1H-pyrazolo[3,4-b]pyridin-6-yl)-9-methoxy-3H-benzo[f]chromen-3-one* (**9h**)

White solid, m.p.: 248–250 °C; IR (KBr, cm^−1^) *ν*: 2968, 1724, 1631, 1573, 1507, 1434, 1414, 1384, 1354, 1281, 1241, 1135, 1105, 980, 960, 905, 827, 789, 758, 705, 692, 636; ^1^H-NMR (400 MHz, DMSO-*d*_6_) *δ* (ppm): 9.32 (s, 1H, ArH), 8.24 (d, *J* = 8.0 Hz, 2H, ArH), 8.17 (d, *J* = 9.2 Hz, 1H, ArH), 8.00–7.97 (m, 2H, ArH), 7.61–7.57 (m, 3H, ArH), 7.51–7.46 (m, 5H, ArH), 7.27–7.24 (m, 2H, ArH), 3.91 (s, 3H, OCH_3_), 2.54–2.53 (m, 2H, CH_2_), 1.89 (s, 3H, CH_3_), 0.86 (t, *J* = 7.6 Hz, 3H, CH_3_); ^13^C-NMR (75 MHz, DMSO-*d*_6_) *δ* (ppm): 160.3, 160.2, 154.5, 154.4, 148.7, 145.5, 142.8, 140.6, 139.5, 135.9, 134.0, 131.5, 131.0, 130.5, 129.6, 129.1, 129.0, 128.9, 127.0, 125.9, 125.7, 120.6, 118.7, 115.8, 114.3, 112.8, 102.7, 56.3, 22.5, 15.8, 14.2; HRMS: *m*/*z* cacld. for C_35_H_28_N_3_O_3_ [M + H]^+^ 538.2131, Found 538.2122.

#### 4.3.13. *2-(5-Ethyl-3-methyl-1-phenyl-4-(p-tolyl)-1H-pyrazolo[3,4-b]pyridin-6-yl)-9-methoxy-3H-benzo[f]chromen-3-one* (**9i**)

Yellow solid, m.p.: >300 °C; IR (KBr, cm^−1^) *ν*: 2966, 1720, 1628, 1570, 1417, 1383, 1264, 1084, 959, 904, 832, 796, 761, 725, 691, 678, 640, 602; ^1^H-NMR (400 MHz, CF_3_COOD) *δ* (ppm): 9.24 (s, 1H, ArH), 8.20 (d, *J* = 9.2 Hz, 1H, ArH), 7.95 (d, *J* = 8.8 Hz, 1H, ArH), 7.65–7.54 (m, 8H, ArH), 7.46 (d, *J* = 9.2 Hz, 1H, ArH), 7.39 (d, *J* = 7.6 Hz, 3H, ArH), 4.04 (s, 3H, OCH_3_), 2.92–2.87 (m, 2H, CH_2_), 2.52 (s, 3H, CH_3_), 2.20 (s, 3H, CH_3_), 1.02 (t, *J* = 7.6 Hz, 3H, CH_3_); ^13^C-NMR (75 MHz, CF_3_COOD) *δ* (ppm): 162.9, 160.1, 155.9, 148.6, 145.6, 145.5, 142.0, 139.8, 137.8, 135.1, 133.4, 131.8, 131.4, 130.6, 130.4, 129.6, 128.9, 127.0, 126.8, 125.7, 121.1, 117.9, 115.3, 113.6, 112.1, 55.3, 22.0, 19.6, 13.5, 11.8; HRMS: *m*/*z* cacld. for C_36_H_30_N_3_O_3_ [M + H]^+^ 552.2287, Found 552.2246.

#### 4.3.14. *2-(5-Ethyl-4-(4-methoxyphenyl)-3-methyl-1-phenyl-1H-pyrazolo[3,4-b]pyridin-6-yl)-9-methoxy-3H-benzo[f]chromen-3-one* (**9j**)

White solid, m.p.: 256–258 °C; IR (KBr, cm^−1^) *ν*: 2965, 2145, 1735, 1717, 1629, 1572, 1463, 1381, 1286, 1227, 1077, 960, 887, 884, 805, 691, 604, 567; ^1^H-NMR (400 MHz, DMSO-*d*_6_) *δ* (ppm): 9.33 (s, 1H, ArH), 8.23 (d, *J* = 8.0 Hz, 2H, ArH), 8.19 (d, *J* = 8.8 Hz, 1H, ArH), 8.01–7.99 (m, 2H, ArH), 7.52–7.48 (m, 3H, ArH), 7.39–7.38 (m, 2H, ArH), 7.28–7.25 (m, 2H, ArH), 7.16 (d, *J* = 8.8 Hz, 2H, ArH), 3.92 (s, 3H, OCH_3_), 2.87 (s, 3H, OCH_3_), 2.56–2.55 (m, 2H, CH_2_), 1.95 (s, 3H, CH_3_), 0.87 (t, *J* = 7.2 Hz, 3H, CH_3_); ^13^C-NMR (75 MHz, DMSO-*d*_6_) *δ* (ppm): 160.3, 160.2, 159.7, 154.6, 154.4, 148.8, 145.5, 142.9, 140.6, 139.6, 134.0, 131.5, 131.0, 130.9, 130.4, 129.7, 127.8, 127.1, 125.9, 125.7, 120.6, 118.8, 116.2, 114.3, 112.8, 102.7, 56.4, 55.7, 22.5, 15.8, 14.5; HRMS: *m*/*z* cacld. for C_36_H_30_N_3_O_4_ [M + H]^+^ 568.2236, Found 568.2248.

#### 4.3.15. *2-(3,5-Dimethyl-1,4-diphenyl-1H-pyrazolo[3,4-b]pyridin-6-yl)-3H-benzo[f]chromen-3-one* (**9k**)

Yellow solid, m.p.: >300 °C; IR (KBr, cm^−1^) *ν*: 2934, 2173, 1710, 1598, 1572, 1438, 1278, 965, 909, 820, 791, 692, 651, 633, 585; ^1^H-NMR (400 MHz, CF_3_COOD) *δ* (ppm): 9.33 (s, 1H, ArH), 8.29–8.24 (m, 2H, ArH), 8.00–7.97 (m, 1H, ArH), 7.72–7.61 (m, 11H, ArH), 7.48–7.47 (m, 2H, ArH), 2.42 (s, 3H, CH_3_), 2.22 (s, 3H, CH_3_); ^13^C-NMR (75 MHz, CF_3_COOD) *δ* (ppm): 155.3, 148.3, 146.2, 145.8, 139.9, 138.5, 133.5, 132.4, 131.8, 131.1, 130.8, 130.4, 130.0, 129.4, 129.3, 129.0, 128.7, 127.7, 126.9, 125.7, 120.6, 120.4, 115.9, 115.5, 112.9, 14.5, 12.0; HRMS: *m*/*z* cacld. for C_33_H_24_N_3_O_2_ [M + H]^+^ 494.1869, Found 494.1887.

#### 4.3.16. *2-(3,5-Dimethyl-1-phenyl-4-(p-tolyl)-1H-pyrazolo[3,4-b]pyridin-6-yl)-3H-benzo[f]chromen-3-one* (**9l**)

Yellow solid, m.p.: 286–290 °C; IR (KBr, cm^−1^) *ν*: 3078, 2187, 1719, 1626, 1606, 1575, 1507, 1447, 1380, 1212, 1093, 963, 813, 790, 741, 685; ^1^H-NMR (400 MHz, CF_3_COOD) *δ* (ppm): 9.35 (s, 1H, ArH), 8.31 (d, *J* = 9.2 Hz, 1H, ArH), 8.27 (d, *J* = 7.6 Hz, 1H, ArH), 8.02 (d, *J* = 8.4 Hz, 1H, ArH), 7.79 (t, *J* = 7.2 Hz, 1H, ArH), 7.72–7.57 (m, 9H, ArH), 7.39 (d, *J* = 7.6 Hz, 2H, ArH), 2.55 (s, 3H, CH_3_), 2.46 (s, 3H, CH_3_), 2.29 (s, 3H, CH_3_); ^13^C-NMR (75 MHz, CF_3_COOD) *δ* (ppm): 154.3, 147.5, 145.2, 144.5, 141.3, 138.9, 137.5, 132.5, 130.8, 130.2, 129.4, 129.0, 128.8, 128.4, 128.3, 128.1, 127.7, 126.7, 126.0, 124.7, 119.6, 119.5, 114.9, 114.4, 18.6, 14.0, 10.9; HRMS: *m*/*z* cacld. for C_34_H_26_N_3_O_2_ [M + H]^+^ 508.2025, Found 508.2020.

#### 4.3.17. *2-(4-(4-Methoxyphenyl)-3,5-dimethyl-1-phenyl-1H-pyrazolo[3,4-b]pyridin-6-yl)-3H-benzo [f]chromen-3-one* (**9m**)

White solid, m.p.: 258–260 °C; IR (KBr, cm^−1^) *ν*: 2904, 2342, 1735, 1631, 1574, 1427, 1367, 1240, 1200, 1158, 1103, 1061, 849, 818, 759, 712, 668, 589; ^1^H-NMR (400 MHz, DMSO-*d*_6_) *δ* (ppm): 9.20 (s, 1H, ArH), 8.66 (d, *J* = 8.4 Hz, 1H, ArH), 8.28 (t, *J* = 7.6 Hz,3H, ArH), 8.11 (d, *J* = 8.0 Hz, 1H, ArH), 7.75–7.63 (m, 3H, ArH), 7.50 (t, *J* = 8.0 Hz, 2H, ArH), 7.37 (d, *J* = 8.4 Hz, 2H, ArH), 7.26 (t, *J* = 7.2 Hz, 1H, ArH), 7.16 (d, *J* = 8.4 Hz, 2H, ArH), 3.87 (s, 3H, OCH_3_), 2.13 (s, 3H, CH_3_), 2.02 (s, 3H, CH_3_); ^13^C-NMR (75 MHz, DMSO-*d*_6_) *δ* (ppm): 159.8, 159.6, 154.2, 154.0, 149.0, 145.3, 142.7, 140.3, 139.6, 134.3, 130.6, 130.5, 129.6 129.5, 128.1, 126.7, 125.8, 124.9, 123.1, 120.6, 117.1, 115.9, 114.4, 113.6, 55.7, 16.3, 14.7; HRMS: *m*/*z* cacld. for C_34_H_26_N_3_O_3_ [M + H]^+^ 524.1974, Found 524.1978.

#### 4.3.18. *2-(4-(3-Methoxyphenyl)-3,5-dimethyl-1-phenyl-1H-pyrazolo[3,4-b]pyridin-6-yl)-3H-benzo[f]chromen-3-one* (**9n**)

White solid, m.p.: 260–263 °C; IR (KBr, cm^−1^) *ν*: 2970, 2372, 1718, 1573, 1505, 1410, 1362, 1279, 1239, 1142, 1054, 1019, 988, 970, 877, 815, 786, 744, 714, 670, 586; ^1^H-NMR (400 MHz, DMSO-*d*_6_) *δ* (ppm): 9.19 (s, 1H, ArH), 8.66 (d, *J* = 8.4 Hz, 1H, ArH), 8.29–8.25 (m, 3H, ArH), 8.10 (d, *J* = 8.0 Hz, 1H, ArH), 7.75–7.63 (m, 3H, ArH), 7.52–7.48 (m, 3H, ArH), 7.25 (t, *J* = 7.6 Hz, 1H, ArH), 7.13–7.11 (m, 1H, ArH), 7.00–6.98 (m, 2H, ArH), 3.84 (s, 3H, OCH_3_), 2.13 (s, 3H, CH_3_), 2.00 (s, 3H, CH_3_); ^13^C-NMR (75 MHz, DMSO-*d*_6_) *δ* (ppm): 159.7, 159.6, 154.3, 154.0, 148.9, 145.2, 142.7, 140.4, 139.6, 137.6, 134.3, 130.5, 130.3, 129.6, 129.5, 129.4, 129.0, 128.1, 126.8, 125.9, 124.5, 123.1, 121.3, 120.6, 117.1, 115.5, 114.7, 114.6, 113.6, 55.8, 16.2, 14.4; HRMS: *m*/*z* cacld. for C_34_H_26_N_3_O_3_ [M + H]^+^ 524.1974, Found 524.1978.

#### 4.3.19. *2-(3,5-Dimethyl-1-phenyl-4-(p-tolyl)-1H-pyrazolo[3,4-b]pyridin-6-yl)-9-methoxy-3H-benzo[f]chromen-3-one* (**9o**)

Yellow solid, m.p.: 288–290 °C; IR (KBr, cm^−1^) *ν*: 2929, 1718, 1631, 1600, 1346, 1239, 1204, 1173, 1149, 1125, 1019, 852, 827, 795, 749, 690, 643, 606; ^1^H-NMR (400 MHz, CF_3_COOD) *δ* (ppm): 9.28 (s, 1H, ArH), 8.22 (d, *J* = 8.8 Hz, 1H, ArH), 7.96 (d, *J* = 8.8 Hz, 1H, ArH), 7.65–7.62 (m, 6H, ArH), 7.56–7.54 (m, 2H, ArH), 7.46 (d, *J* = 8.8 Hz, 1H, ArH), 7.41–7.35 (m, 3H, ArH), 4.04 (s, 3H, OCH_3_), 2.52 (s, 3H, CH_3_), 2.44 (s, 3H, CH_3_), 2.26 (s, 3H, CH_3_); ^13C-NMR (75 MHz, CF^_3_COOD) *δ* (ppm): 159.0, 158.9, 155.0, 147.4, 145.2, 144.4, 141.2, 138.8, 137.0, 132.4, 130.7, 130.4, 129.6, 129.4, 128.8, 128.2, 128.1, 126.0, 125.8, 124.6, 119.4, 116.6, 115.2, 114.1, 112.5, 54.2, 18.5, 14.0, 10.9; HRMS: *m*/*z* cacld. for C_35_H_28_N_3_O_3_ [M + H]^+^ 538.2131, Found 538.2130.

#### 4.3.20. *9-Methoxy-2-(4-(4-methoxyphenyl)-3,5-dimethyl-1-phenyl-1H-pyrazolo[3,4-b]pyridin-6-yl)-3H-benzo[f]chromen-3-one* (**9p**)

Yellow solid, m.p.: 287–289 °C; IR (KBr, cm^−1^) *ν*: 1716, 1630, 1611, 1571, 1513, 1464, 1385, 1246, 1107, 1033, 960, 832, 795, 754, 691; ^1^H-NMR (400 MHz, CF_3_COOD) *δ* (ppm): 9.29 (s, 1H, ArH), 8.23 (d, *J* = 8.8 Hz, 1H, ArH), 7.56 (d, *J* = 8.8 Hz, 1H, ArH), 7.65–7.63 (m, 6H, ArH), 7.48–7.42 (m, 3H, ArH), 7.39–7.35 (m, 3H, ArH), 4.07–4.05 (m, 6H, 2 × OCH_3_), 2.46 (s, 3H, CH_3_), 2.30 (s, 3H, CH_3_); ^13C-NMR (75 MHz, CF^_3_COOD) *δ* (ppm): 159.6, 159.0, 155.0, 147.2, 145.1, 144.6, 138.9, 137.0, 132.4, 130.7, 130.4, 129.6, 129.3, 128.3, 128.1, 125.8, 124.6, 116.6, 114.0, 112.5, 54.3, 54.0, 14.0, 11.1; HRMS: *m*/*z* cacld. for C_35_H_28_N_3_O_4_ [M + H]^+^ 554.2080, Found 554.2093.

#### 4.3.21. *2-(3,5-Dimethyl-1,4-diphenyl-1H-pyrazolo[3,4-b]pyridin-6-yl)-9-methoxy-3H-benzo[f]chromen-3-one* (**9q**)

Yellow solid, m.p.: 252–254 °C; IR (KBr, cm^−1^) *ν*: 2961, 1725, 1629, 1582, 1557, 1435, 1397, 1335, 1290, 1250, 1219, 1196, 999, 906, 819, 797, 753, 695, 625; ^1^H-NMR (400 MHz, DMSO-*d*_6_) *δ* (ppm): 9.29 (s, 1H, ArH), 8.28–8.26 (m, 2H, ArH), 8.17 (d, *J* = 9.2 Hz, 1H, ArH), 8.01–7.98 (m, 2H, ArH), 7.62–7.57 (m, 3H, ArH), 7.52–7.43 (m, 5H, ArH), 7.28–7.25 (m, 2H, ArH), 3.92 (s, 3H, OCH_3_), 2.11 (s, 3H, CH_3_), 1.95 (s, 3H, CH_3_); ^13^C-NMR (75 MHz, DMSO-*d*_6_) *δ* (ppm): 160.2, 159.7, 154.6, 154.5, 148.9, 145.3, 142.6, 140.8, 139.6, 136.2, 134.0, 131.5, 131.1, 129.7, 129.2, 129.1, 127.4, 125.9, 125.7, 124.5, 120.7, 118.7, 115.5, 114.3, 112.9, 102.8, 56.3, 16.2, 14.5; HRMS: *m*/*z* cacld. for C_34_H_26_N_3_O_3_ [M + H]^+^ 524.1974, Found 524.1988.

#### 4.3.22. *2-(3-Methyl-1,4-diphenyl-1H-pyrazolo[3,4-b]pyridin-6-yl)-3H-benzo[f]chromen-3-one* (**9r**)

Yellow solid, m.p.: 268–270 °C; IR (KBr, cm^−1^) *ν*: 2935, 2355, 1729, 1667, 1553, 1092, 891, 818, 746, 694, 657, 631, 585; ^1^H-NMR (400 MHz, CF_3_COOD) *δ* (ppm): 10.22 (s, 1H, ArH), 8.61–8.57 (m, 2H, ArH), 8.46 (d, *J* = 9.2 Hz, 1H, ArH), 8.09 (d, *J* = 8.0 Hz, 1H, ArH), 7.90 (t, *J* = 7.6 Hz, 1H, ArH), 7.82–7.76 (m, 11H, ArH), 7.70 (d, *J* = 8.8 Hz, 1H, ArH), 2.58 (s, 3H, CH_3_); ^13C-NMR (75 MHz, CF^_3_COOD) *δ* (ppm): 163.5, 159.1, 155.0, 147.5, 145.0, 144.8, 140.3, 138.7, 132.9, 132.4, 131.1, 130.4, 130.3, 130.0, 129.8, 128.8, 128.3, 127.9, 127.4, 127.3, 122.6, 120.1, 114.3, 114.1, 113.5, 11.9; HRMS: *m*/*z* cacld. for C_32_H_22_N_3_O_2_ [M + H]^+^ 480.1712, Found 480.1726.

#### 4.3.23. *2-(4-(4-Methoxyphenyl)-3-methyl-1-phenyl-1H-pyrazolo[3,4-b]pyridin-6-yl)-3H-benzo[f]chromen-3-one* (**9s**)

Yellow solid, m.p.: >300 °C; IR (KBr, cm^−1^) *ν*: 2988, 2355, 1987, 1730, 1512, 1089, 1066, 959, 810, 809, 788, 765, 689, 654, 633, 599; ^1^H-NMR (400 MHz, CF_3_COOD) *δ* (ppm): 11.03 (s, 1H, ArH), 9.43 (d, *J* = 8.4 Hz, 1H, ArH), 9.34–9.30 (m, 2H, ArH), 8.94 (d, *J* = 8.0 Hz, 1H, ArH), 8.75 (t, *J* = 7.6 Hz, 1H, ArH), 7.66–7.63 (m, 8H, ArH), 8.54 (t, *J* = 9.2 Hz, 1H, ArH), 8.23 (d, *J* = 8.8 Hz, 2H, ArH), 4.95 (s, 3H, OCH_3_), 3.50 (s, 3H, CH_3_); ^13C-NMR (75 MHz, CF^_3_COOD) *δ* (ppm): 164.3, 159.2, 155.9, 148.2, 145.7, 145.3, 141.1, 139.8, 133.6, 131.3, 130.9, 130.6, 129.7, 128.8, 128.2, 126.6, 123.5, 120.9, 117.1, 115.1, 114.6, 55.1, 13.1; HRMS: *m*/*z* cacld. for C_33_H_24_N_3_O_3_ [M + H]^+^ 510.1818, Found 510.1835.

#### 4.3.24. *2-(5-Ethyl-1-methyl-3,4-diphenyl-1H-pyrazolo[3,4-b]pyridin-6-yl)-3H-benzo[f]chromen-3-one* (**9t**)

Yellow solid, m.p.: 285–288 °C; IR (KBr, cm^−1^) *ν*: 2396, 1732, 1574, 1353, 1099, 1515, 1088, 1076, 959, 810, 803, 704; ^1^H-NMR (400 MHz, CF_3_COOD) *δ* (ppm): 10.26 (s, 1H, ArH), 9.25 (d, *J* = 8.8 Hz, 2H, ArH), 8.95 (d, *J* = 8.0 Hz, 1H, ArH), 8.73 (t, *J* = 7.6 Hz, 1H, ArH), 8.63 (t, *J* = 7.6 Hz, 1H, ArH), 8.46 (d, *J* = 7.6 Hz, 1H, ArH), 8.26 (t, *J* = 7.6 Hz, 1H, ArH), 8.21–8.14 (m, 3H, ArH), 8.09–8.02 (m, 4H, ArH), 7.94 (d, *J* = 7.6 Hz, 2H, ArH), 5.31 (s, 3H, CH_3_), 3.87–3.85 (m, 2H, CH_2_), 1.84 (t, *J* = 7.2 Hz, 3H, CH_3_); ^13C-NMR (75 MHz, CF^_3_COOD) *δ* (ppm): 162.9, 155.3, 150.3, 146.0, 145.4, 140.3, 138.2, 134.3, 131.2, 131.1, 130.2, 129.9, 129.5, 129.3, 128.5, 128.3, 127.9, 127.8, 127.6, 127.1, 120.5, 119.3, 116.3, 115.5, 34.9, 21.8, 13.1; HRMS: *m*/*z* cacld. for C_34_H_26_N_3_O_2_ [M + H]^+^ 508.2025, Found 508.2027.

#### 4.3.25. *2-(1,5-Dimethyl-3,4-diphenyl-1H-pyrazolo[3,4-b]pyridin-6-yl)-9-methoxy-3H-benzo[f]chromen-3-one* (**9u**)

Yellow solid, m.p.: 260–262 °C; IR (KBr, cm^−1^) *ν*: 2697, 2551, 1783, 1708, 1628, 1567, 1511, 1469, 1441, 1387, 1330, 1218, 1149, 1017, 976, 898, 845, 796, 756, 725, 702, 601; ^1^H-NMR (400 MHz, CF_3_COOD) *δ* (ppm): 10.10 (s, 1H, ArH), 9.03 (d, *J* = 9.2 Hz, 1H, ArH), 8.76 (d, *J* = 8.8 Hz, 1H, ArH), 8.66–8.65 (m, 1H, ArH), 8.28 (d, *J* = 8.8 Hz, 1H, ArH), 8.20 (d, *J* = 9.2 Hz, 1H, ArH), 8.12 (d, *J* = 7.6 Hz, 1H, ArH), 8.05–7.99 (m, 3H, ArH), 7.91–7.87 (m, 4H, ArH), 7.81 (d, *J* = 7.6 Hz, 2H, ArH), 5.18 (s, 3H, OCH_3_), 4.85 (s, 3H, CH_3_), 3.21 (s, 3H, CH_3_). ^13C-NMR (75 MHz, CF^_3_COOD) *δ* (ppm): 162.3, 160.0, 156.0, 150.1, 146.1, 145.8, 140.3, 138.0, 131.5, 131.4, 130.6, 130.3, 129.5, 128.5, 128.3, 128.2, 128.0, 127.9, 127.2, 126.7, 117.5, 113.5, 112.2, 112.1, 103.0, 55.2, 34.8, 15.1. HRMS: *m*/*z* cacld. for C_34_H_26_N_3_O_3_ [M + H]^+^ 554.2080, Found 554.2093.

#### 4.3.26. *9-Methoxy-2-(4-(4-methoxyphenyl)-1,5-dimethyl-3-phenyl-1H-pyrazolo[3,4-b]pyridin-6-yl)-3H-benzo[f]chromen-3-one* (**9v**)

Yellow solid, m.p.: 240–244 °C; IR (KBr, cm^−1^) *ν*: 2932, 1720, 1624, 1608, 1564, 1512, 1463, 1383, 1353, 1289, 1249, 1208, 1173, 1025, 970, 902, 836, 801, 698, 664, 607; ^1^H-NMR (400 MHz, CF_3_COOD) *δ* (ppm): 10.15 (s, 1H, ArH), 9.09 (d, *J* = 9.2 Hz, 1H, ArH), 8.82 (d, *J* = 9.2 Hz, 1H, ArH), 8.72–8.71 (m, 1H, ArH), 8.34 (d, *J* = 9.2 Hz, 1H, ArH), 8.26 (d, *J* = 9.2 Hz, 1H, ArH), 8.17–8.13 (m, 1H, ArH), 8.02–7.95 (m, 4H, ArH), 7.89–7.87 (m, 2H, ArH), 7.69–7.66 (m, 2H, ArH), 5.23 (s, 3H, OCH_3_), 4.90 (s, 3H, OCH_3_), 4.72 (s, 3H, CH_3_), 3.31 (s, 3H, CH_3_); ^13C-NMR (75 MHz, CF^_3_COOD) *δ* (ppm): 162.4, 160.3, 156.1, 150.0, 146.0, 145.9, 140.3, 138.1, 131.5, 130.4, 129.4, 128.7, 128.1, 128.0, 127.6, 126.8, 125.1, 117.5, 114.2, 113.5, 55.2, 55.1, 34.9, 15.1; HRMS: *m*/*z* cacld. for C_35_H_28_N_3_O_4_ [M + H]^+^ 524.1974, Found 524.1972.

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
