# Peer review of "Alcohol Participates in the Synthesis of Functionalized Coumarin-Fused Pyrazolo[3,4-*b*]Pyridine from a One-Pot Three-Component Reaction"

_molecules, 2019, doi:10.3390/molecules24152835_

Round 1
Reviewer 1 Report
The manuscript by Wang et al. reports a novel three-component reaction that leads to coumarin-fused pyrazolo[3,4-b]pyridines – potential biologically active compounds. The reaction involves an alcohol that generates an aldehyde using SSA as the catalyst according to the mechanism proposed by authors. Scope of the reaction does have limitations on substituents but it is quite broad, therefore this new method is of potential interest to readers of Molecules and medicinal chemists.
The authors need to address some concerns before the manuscript can be considered for publication. First of all, they must provide Supporting Information that is missing in their original submission. There are other concerns:
1) Figure 1: Structures of columbianetin, praeruptorin A and marmesin are wrong. Check and correct them.
2) Figure 2: Drug #5 must be anxiolytic.
3) How confident are the authors in steric effect of benzocoumarin vs. coumarin? I did not see the reaction with coumarins performed in EG/EtOH 3:1. What would be the outcome?
4) Don’t call benzyl alcohols as aryl alcohols.
5) Lines 97-98: The sentence does not make sense: CH3 cannot be connected to the electrons, and the reaction did not proceed successfully.
6) Scheme 4: Structure of 9a appears to be incorrect. What was the oxidant when benzyl alcohol was subjected to MWI using SSA? Was it oxygen from air? Will reaction occur with benzaldehyde without SSA?
7) Scheme 5: Specify the oxidant that generates the aldehyde. A control experiment with benzaldehyde without SSA needed to confirm the necessity of SSA in the entire process (not only for generation of an aldehyde from an alcohol).
I would be happy to reconsider a revised manuscript after addressing these concerns and revising text for grammatical and typing errors.
Author Response
Response to Reviewer 1 Comments
Dear Editor,
Thank the reviewers for giving so many valuable and pertinent suggestions. In light of reviewers’ advice, we have carefully revised the manuscript indicated below.
Point 1: Figure 1: Structures of columbianetin, praeruptorin A and marmesin are wrong. Check and correct them.
Response 1: According to the reviewer’s suggestion, the structures of columbianetin, praeruptorin A and marmesin have been changed in the revised manuscript.
Point2: Figure 2: Drug #5 must be anxiolytic.
Response 2: Thanks! It was revised according to reviewer’s suggestion.
Point 3: How confident are the authors in steric effect of benzocoumarin vs. coumarin? I did not see the reaction with coumarins performed in EG/EtOH 3:1. What would be the outcome?
Response 3: In the early literature reports of our group, the coumarino[4,3-d] pyrazolo[3,4-b]pyridine derivative (3a) was synthesized. Other solvents(including EG/EtOH 3:1) and reaction temperatures we have also tried, but the product only 3a and no target product was produced.
Point 4: Don’t call benzyl alcohols as aryl alcohols.
Response 4: Thanks! It was revised according to reviewer’s suggestion, please see revised manuscript.
Point 5: Lines 97-98: The sentence does not make sense: CH3 cannot be connected to the electrons, and the reaction did not proceed successfully.
Response 5: Thanks for your advice. It was revised according to reviewer’s suggestion.
Point 6: Scheme 4: Structure of 9a appears to be incorrect. What was the oxidant when benzyl alcohol was subjected to MWI using SSA? Was it oxygen from air? Will reaction occur with benzaldehyde without SSA?
Response 1: According to the reviewer’s suggestion, the structures of 9a has been changed in the revised manuscript. The oxidant might be the air in the reactor. We have using arylbenzo[f]chromen-3-one 4, enaminone 2 and benzaldehyde 10 as staring materials without SSA catalyzed three-component domino reaction under microwave irradiation, the reaction did not proceed successfully。
Point 7: Scheme 5: Specify the oxidant that generates the aldehyde. A control experiment with benzaldehyde without SSA needed to confirm the necessity of SSA in the entire process (not only for generation of an aldehyde from an alcohol).
Response 7: Thanks for your advice, the control experiment using arylbenzo[f] chromen-3-one 4, enaminone 2 and benzaldehyde 10 as staring materials three-component domino reaction under microwave irradiation without SSA have been tried, and the reaction did not proceed successfully.
Response to Reviewer 1 Comments
Dear Editor,
Thank the reviewers for giving so many valuable and pertinent suggestions. In light of reviewers’ advice, we have carefully revised the manuscript indicated below.
Point 1: Figure 1: Structures of columbianetin, praeruptorin A and marmesin are wrong. Check and correct them.
Response 1: According to the reviewer’s suggestion, the structures of columbianetin, praeruptorin A and marmesin have been changed in the revised manuscript.
Point2: Figure 2: Drug #5 must be anxiolytic.
Response 2: Thanks! It was revised according to reviewer’s suggestion.
Point 3: How confident are the authors in steric effect of benzocoumarin vs. coumarin? I did not see the reaction with coumarins performed in EG/EtOH 3:1. What would be the outcome?
Response 3: In the early literature reports of our group, the coumarino[4,3-d] pyrazolo[3,4-b]pyridine derivative (3a) was synthesized. Other solvents(including EG/EtOH 3:1) and reaction temperatures we have also tried, but the product only 3a and no target product was produced.
Point 4: Don’t call benzyl alcohols as aryl alcohols.
Response 4: Thanks! It was revised according to reviewer’s suggestion, please see revised manuscript.
Point 5: Lines 97-98: The sentence does not make sense: CH3 cannot be connected to the electrons, and the reaction did not proceed successfully.
Response 5: Thanks for your advice. It was revised according to reviewer’s suggestion.
Point 6: Scheme 4: Structure of 9a appears to be incorrect. What was the oxidant when benzyl alcohol was subjected to MWI using SSA? Was it oxygen from air? Will reaction occur with benzaldehyde without SSA?
Response 1: According to the reviewer’s suggestion, the structures of 9a has been changed in the revised manuscript. The oxidant might be the air in the reactor. We have using arylbenzo[f]chromen-3-one 4, enaminone 2 and benzaldehyde 10 as staring materials without SSA catalyzed three-component domino reaction under microwave irradiation, the reaction did not proceed successfully。
Point 7: Scheme 5: Specify the oxidant that generates the aldehyde. A control experiment with benzaldehyde without SSA needed to confirm the necessity of SSA in the entire process (not only for generation of an aldehyde from an alcohol).
Response 7: Thanks for your advice, the control experiment using arylbenzo[f] chromen-3-one 4, enaminone 2 and benzaldehyde 10 as staring materials three-component domino reaction under microwave irradiation without SSA have been tried, and the reaction did not proceed successfully.

Reviewer 2 Report
J. Wang and coworkers described a new method for the preparation of substituted coumarins. The possibility to obtain modified coumarins in short time and using a simple, one-pot methodology is of scientific interest due to the many potential applications of the coumarin scaffold. However, the manuscript is often difficult to read and the language should be revised. The experimental procedures are detailed and the compounds well characterized (it would have been nice to have the spectra available for comparison).
Several open questions remain after reading the article. First and most important: if the aldehyde is most probably the reactive species in the reaction, why not to use it as reagent instead of the alcohol? The authors already showed that the reaction is much faster when the aldehyde is used. Moreover, a larger variety of alcohols (aldehydes) should be tested to demonstrate that the reaction is not limited to few particular cases. When using the aldehyde, would the reaction work also without microwave irradiation? Under non-optimal conditions, e.g., solvent = EG/EtOH 1:2, can other products be identified?
Author Response
Response to Reviewer 2 Comments
Dear Editor,
Thank the reviewers for giving so many valuable and pertinent suggestions. In light of reviewers’ advice, we have carefully revised the manuscript indicated below.
Point 1: The manuscript is often difficult to read and the language should be revised. The experimental procedures are detailed and the compounds well characterized (it would have been nice to have the spectra available for comparison).
Response 1: According to the reviewer’s suggestion, in the revised manuscript, the language and some spelling errors have been corrected. And the supplementary materials (Crystal date of compound 7a, 1H NMR and 13C NMR Spectra of all compounds and GC-MS spectra of Scheme 4B) have been provided
Point 2: Several open questions remain after reading the article. First and most important: if the aldehyde is most probably the reactive species in the reaction, why not to use it as reagent instead of the alcohol? The authors already showed that the reaction is much faster when the aldehyde is used. Moreover, a larger variety of alcohols (aldehydes) should be tested to demonstrate that the reaction is not limited to few particular cases. When using the aldehyde, would the reaction work also without microwave irradiation? Under non-optimal conditions, e.g., solvent = EG/EtOH 1:2, can other products be identified?
Response 2: Thanks for your advice, aldehyde can promote this reaction, but we consider is more meaningful that alcohols perform in this reaction, the reaction goes through the steps of the oxidation reaction. When using the aldehyde as starting material, the reaction can proceed smoothly under microwave irradiation conditions, but the yield target product is very low under heating conditions. The best conditions for this reaction (temperature, solvent, catalyst, etc.) have been changed, and this part is under studying.

Round 2
Reviewer 1 Report
The authors have addressed my comments and provided supporting information with all experimental evidence. The manuscript is now suitable for publication after minor revisions noted:
1) There must be oxygen instead of methylene unit in the structure of Praeruptorin A.
2) “Results and discussion” and all compound numbers in the experimental section must be bold.
3) Scheme 4B. Add “air” above the reaction arrow.
4) Text of the manuscript was improved but I highly recommend to go through it again thoroughly and check for errors.
Author Response
Response to Reviewer 1 Comments
Dear Editor,
Thank the reviewers for giving so many valuable and pertinent suggestions. In light of reviewers’ advice, we have carefully revised the manuscript indicated below.
Point 1: There must be oxygen instead of methylene unit in the structure of Praeruptorin A
Response 1: According to the reviewer’s suggestion, the structure of praeruptorin A has been changed in the revised manuscript.
Point 2: “Results and discussion” and all compound numbers in the experimental section must be bold.
Response 2: Thanks! “Results and discussion” and all compound numbers in the experimental section have been bold. Please see the manuscript.
Point 3: Scheme 4B. Add “air” above the reaction arrow.
Response 3: According to the reviewer’s suggestion, it was revised, please see the manuscript.
Point 4: Text of the manuscript was improved but I highly recommend to go through it again thoroughly and check for errors.
Response 4: Thank you very much for your suggestion. We have checked the manuscript and revised it according to the comments.

Reviewer 2 Report
The revised version of the manuscript from Wang and coworkers now includes supplementary material with the experimental data of the described compounds, showing that the new coumarins have been all obtained in good purity and that the spectral data enabled convincing structural assignement. No further improvement compared to the previous version has been performed, in particular concerning the variety of alcohols that could be used, as well as comparison with the corresponding aldehydes, which would eventually prove the effective convenience of using the alcohol instead of the aldehyde.
I anyway believe that the presented results are of interest for the scientific community, as they show the preparation of quite complex molecules in a one pot way, for this reason and considering what discussed above I would recommend publication in a more specialize journal.
Author Response
Response to Reviewer 2 Comments
Dear Editor,
Thank the reviewers for giving so many valuable and pertinent suggestions. In light of reviewers’ advice, we have carefully revised the manuscript indicated below.
The revised version of the manuscript from Wang and coworkers now includes supplementary material with the experimental data of the described compounds, showing that the new coumarins have been all obtained in good purity and that the spectral data enabled convincing structural assignement. No further improvement compared to the previous version has been performed, in particular concerning the variety of alcohols that could be used, as well as comparison with the corresponding aldehydes, which would eventually prove the effective convenience of using the alcohol instead of the aldehyde.
Response: Thanks! According to the reviewer’s suggestion, the (4-bromophenyl)methanol, pyridin-4-ylmethanol and furan-2-ylmethanol have been used as the starting materials, the corresponding products were afforded.
For this reaction, alcohol adapts more widely than aldehydes, aromatic aldehydes can be used as raw materials, but when alkyl aldehydes are used instead, the reaction can not proceed smoothly or the yield is low comparison with the corresponding alcohols. When benzaldehyde (10) was added to the reaction instead of phenylmethanol (8a) under standard conditions, 73% yield of desired product (9a) could be obtained. Alkyl aldehyde (acetaldehyde solution) is used as raw materials, the reaction can not proceed smoothly. And when butyraldehyde (11) was added to the reaction 50% yield of desired product (7c) could be obtained. As a reaction material, alcohol has the advantage of being stable, easy to obtain, and cheap compared to aldehydes, so we not using the alcohol instead of the aldehyde.
